# Effects of Long-Term Supplementation of Bovine Colostrum on the Immune System in Young Female Basketball Players. Randomized Trial

**DOI:** 10.3390/nu13010118

**Published:** 2020-12-30

**Authors:** Anna Skarpańska-Stejnborn, Mirosława Cieślicka, Hanna Dziewiecka, Sławomir Kujawski, Anita Marcinkiewicz, Jerzy Trzeciak, Piotr Basta, Dariusz Maciejewski, Ewa Latour

**Affiliations:** 1Department of Biological Sciences, Faculty of Physical Culture in Gorzow Wielkopolski, Poznan University of Physical Education, Estkowskiego 13, 66-400 Gorzów Wielkopolski, Poland; hannadziewiecka@gmail.com; 2Department of Physiology, Collegium Medicum in Bydgoszcz, Nicolaus Copernicus University in Toruń, M. Sklodowskiej-Curie 9, 85-094 Bydgoszcz, Poland; cudaki@op.pl; 3Department of Hygiene, Epidemiology, Ergonomy and Postgraduate Education, Ludwik Rydygier Collegium Medicum in Bydgoszcz Nicolaus Copernicus University in Torun, M. Sklodowskiej-Curie 9, 85-094 Bydgoszcz, Poland; skujawski@cm.umk.pl; 4Central Research Laboratory, Faculty of Physical Culture in Gorzow Wielkopolski, Poznan University of Physical Education, Estkowskiego 13, 66-400 Gorzów Wielkopolski, Poland; malucykbuziak@op.pl (A.M.); jurektrzeciak@o2.pl (J.T.); 5Department of Physical Education and Sport, Faculty of Physical Culture in Gorzow Wielkopolski, Poznan University of Physical Education, Estkowskiego 13, 66-400 Gorzów Wielkopolski, Poland; maly197@interia.pl (P.B.); d.maciejewski@awf-gorzow.edu.pl (D.M.); 6Department of Physiotherapy, Faculty of Physical Culture in Gorzow Wielkopolski, Poznan University of Physical Education, Estkowskiego 13, 66-400 Gorzów Wielkopolski, Poland; ewalatour@o2.pl

**Keywords:** supplementation, physical exercise, sport, bovine colostrum, immunologic, athletes

## Abstract

An intensive physical exercise program could lead to a decrease in immune system function. Effects of long-term supplementation of bovine colostrum on the response of immune function on physical exercise test in athletes were examined. Twenty-seven elite female basketball players (age 16–19) were randomly assigned to either an experimental group or a control group. Eventually, *n* = 11 athletes completed intervention in the experimental group (3.2 g bovine colostrum orally twice a day for 24 weeks), while *n* = 9 athletes in the control group were given a placebo. Before the supplementation, after 3 and 6 months, subjects performed the physical exercise stress test. Before, just after, and 3 h after physical exercise testing, blood was drawn and immune system indicators were examined. Plasma interleukin (IL)-1alpha, IL-2, IL-10, IL-13, tumor necrosis factor (TNF) alpha, creatine kinase (CK MM), immunoglobulin G (IgG), insulin-like growth factor 1 (IGF1), and WBC, lymphocyte (LYM), monocyte (MON), and granulocyte (GRA) were measured. A statistically significant change in IL-10 in response to the exercise program during the supplementation period in both groups was observed (*p* = 0.01). However, the results of the rest of the comparisons were statistically insignificant (*p* > 0.05). Contrary to our initial hypothesis, there were no significant effects of bovine supplementation on the dynamics of immune system function indicators.

## 1. Introduction

Athletes practicing competitive sports are much more prone to depression of the immune system compared to that in people practicing sports recreationally [1]. The phenomenon of exercise-induced immunosuppression is most often observed during intense or prolonged exercise or sports competitions [2]. Glesson et al. estimated that diseases of the upper respiratory tract in athletes account for 35% to 65% of all cases of illnesses not related to sports injuries. Drew et al. [3] conducted a survey of 211 athletes participating in the 2016 Rio Olympics. The authors noticed that female sex, low energy availability, and mental health were associated with sports incapacity due to illness (an illness was defined as an event that limited training or competition for greater hours in the prior month).

The immune system maintains a relative balance that is responsible for the homeostasis of the organism. The antagonism between the cellular and humoral responses in the body underlies a phenomenon known as immunological polarization [4]. T helper type 1 (Th1) and Th2 helper cells mediate cellular and humoral immunity, respectively, and disturbance of Th1/Th2 balance leads to various immune abnormalities. Th1/Th2 balance is important for maintaining the immune health of the host [5]. Th1 cells promote resistance to intracellular pathogens and secrete cytokines related to, inter alia, host viral response. In contrast, Th2 cells promote response to extracellular pathogens and antagonize the production and activity of Th1 cytokines [6]. Thus, an altered Th1/Th2 balance may affect the host’s susceptibility to various immune-mediated diseases including allergy, autoimmunity, and an increased number of infections [7,8]. Maximum intensity physical effort, even in elite athletes, is an extreme stress factor that can cause damage not only to the muscular system but also to other tissues. In response to this type of damage, an organism’s “response” can be observed, which is expressed both by an increase in pro- and anti-inflammatory factors. In this very dynamic system, compensatory mechanisms appear, which aim at its stabilization. However, the maintenance of relative stabilization in the immune system may not be sufficient and, according to many authors, the equilibrium is shifted toward Th2 [9,10,11]. As a result of cellular immunosuppression, players are more susceptible to infection and, therefore, it may result in both increased susceptibility of athletes to diseases (especially the upper respiratory tract infection (URTI)) and an increase in inflammation and symptoms of overtraining [12,13]. The above changes can be observed primarily in the period of increased exercise load [2].

Sports competitions, especially those conducted at the highest level, require a huge amount of work from players. Basketball season lasts relatively long (about 8 months). Significant stresses faced by the players during this period (about 6 matches a month), exhausting travels (half of the games take place outside the place of residence), and the related sleep disturbances and stress accompanying sports competitions are important factors that may affect immunity [14]. Breaks in training due to illness may not only cause a temporary cessation of the training program but also exclude the competitor from participation in the competition, thus destroying his/her many years of preparation. Therefore, it seems advisable to find a method that could limit the effects of such a high effort load and limit the adverse changes observed after exercise. 

Introducing bovine colostrum (BC), with documented immunomodulatory activity [15], into the diet may be one of the elements of a gentle and safe intervention to restore the balance of the immune system. There is convincing evidence that daily supplementation with BC can maintain the integrity of the intestinal barrier, immune function, and reduce the risk of symptoms of URTIs [16,17]. To the best of our knowledge, no work has been published so far that would concern the use of BC supplementation in athletes in team sports and was analyzed in terms of its ability to reduce post-exercise immune suppression. It should be emphasized that BC is a supplement with a high safety profile, and the presence of immunologically active compounds may have a beneficial effect on the immune system in people of all ages. Colostrum in its composition contains lactoferrin, α-lactalbumin, lysozyme, lactoperoxidase, proline-rich polypeptide, and casein. Its composition also shows the presence of immunoglobulin (Ig)A, IgD, IgE, IgG, and IgM [18,19,20]. It is recommended to introduce colostrum into the diet of athletes as a factor that may have a significant impact on reducing diseases resulting from increased training loads [2,21].

The aim of the research was to analyze the effect of long-term supplementation with BC on exercise-induced changes in response to immune system function to physical exercise testing in elite basketball players. BC is presumably an effective and safe therapy to reduce the harmful effects of intense physical effort by reducing exercise-induced immune dysfunction, reducing the level of inflammatory markers, and post-exercise damage of muscle fibers. Therefore, the following research hypothesis was formulated: supplementation with colostrum, by limiting inflammation and reducing damage to muscle fibers induced by intense effort, will have a positive effect on the immune system.

## 2. Materials and Methods

### 2.1. Participants

The research was conducted in the period from April to October (2018) on the players of youth groups of ENEA AZS AJP Gorzów Wlkp. (1st league and extra class). As the World Anti-Doping Agency (WADA) included colostrum on the list of substances that are not recommended by this organization because they may cause an increase in insulin-like growth factor (IGF) levels, the research was conducted outside the launch period. The first test date (April) was at the end of the starting period (the players finished the games). The second test (June/July) was carried out during the rest period of the players from training (the so-called decay period), while the last date (October) fell on the period of immediate preparation for the competition, which was characterized by the highest effort load for the competitors. Due to the possibility of an increase in IGF levels, as a result of the supplementation used in the competitors, the tests were carried out outside the competition period. All the players were one team and were subject to the same training regime, namely: training sessions took place every day in the afternoon and lasted from 1.5 to 2 h. Experimental procedures and potential risks were discussed with the participants, and informed consent forms were provided and signed prior to inclusion in the study. The study was conducted in accordance with the Declaration of Helsinki, and its protocol was approved by the local Ethics Committee at Poznan University of Medical Sciences (Decision no. 714/18 of 14 June 2017).

Sample size calculation was done using an online calculator for parallel study with quantitative measurement [http://hedwig.mgh.harvard.edu/sample_size/js/js_parallel_quant.html], assuming 0.05 as alpha level and 0.8 power. Based on previous research on effects of bovine colostrum on lymphocytes, the difference in mean was assumed as 0.6 and 0.5 as SD of parameter [22]. The ratio of allocation was set at 0.92 due to the possible dropouts in the supplemented group. A total of 24 patients was indicated as a result of this calculation. To adjust for circa 10% of dropout rate in total, 27 participants we assigned to randomization. 

### 2.2. Diet Supplementation

Before the supplementation, the players were randomly divided into two groups. The supplemented group (*n* = 11) received four capsules of BC (produced by AGRAPAK, Poland) every morning and evening. One gel capsule contained 0.4 g of colostrum. The composition of the supplement per single dose of 3.2 g (four capsules): total protein-2.620 g, lactose-0.16 g, fat-0.05 g, active protein substances (lactoferrin-30 mg, Platelet-rich plasma (PRP)-0.16 g, IgG-1050 mg, IGF-16 µg, LZM-21.2 mg, and αLA-30 mg). The PRP content was estimated by measuring the content and ratio of amino acids (Pro and Val) based on the conducted research and analysis of bibliographic data [23]. The placebo group (*n* = 9) in the same dose, form, and date of the competitors received powdered milk. The composition of the placebo calculated for a single dose of 3.2 g: lactose 1.6 g, protein 1.08 g, fat 0.04 g, and ash 0.25. The supplementation period lasted for 24 weeks in total. 

### 2.3. Physical Performance Examination

Before the supplementation, after 3 and 6 months, all the players performed the maximum stress test on the HP Cosmos treadmill. During the test, the aerobic capacity of the participants was assessed. The test protocol was as follows: the starting speed of the treadmill for the runners was 8.0 km/h, then it was increased every 2 min by 1.0 km/h, until exhaustion. Participants were verbally encouraged to continue as long as possible. Heart rate (bpm) was recorded with a sport tester (Polar PE 3000). The data are presented in Table 1. 

### 2.4. Material Collection and Examination

In each of the time points, before the exercise, in the first minute after the end of testing and 3 h of restitution, blood was collected from the participants from the antecubital vein. Blood samples were collected in tubes with dipotassium ethylene diamine tetra-acetic acid (K2EDTA) as an anticoagulant. The list of analyzed parameters included white blood cell (WBC) counts and percentages of lymphocytes and granulocytes, all determined with the MYTHIC 18 Haematology Analyser (Orphee Medical, Geneva, Switzerland).

Plasma interleukin 1 alpha (IL-1alpha), interleukin-2 (IL-2), interleukin-10 (IL-10), interleukin-13 (IL-13), Tumour Necrosis Factor alpha (TNF alpha), creatine kinase (CK MM), immunoglobulin G (IgG), insulin-like growth factor 1 (IGF-1) and white blood cells (WBC), lymphocytes (LYM), monocytes (MON) and granulocytes (GRA). The following tests were used for a detailed analysis of changes in levels IgG (DRG), IGF-1 ((DRG), CK (SunRed), interleukin (IL)- 1α (SunRed), IL-2 (SunRed), IL-10 (SunRed), IL-13 (SunRed), and tumor necrosis factor (TNF)-α. The Thermo Scientific Multiscan GO Microplate Spectrophotometer produced by Fisher Scientific Finland was used for the material examination.

### 2.5. Statistical Analysis

To assess the impact of supplementation on examined factors, a linear mixed model with restricted maximum likelihood approach and t tests using the Satterthwaite method using R statistical packages (Lme4 and LmerTest) were used [24]. The subject factor was determined as a random effect. Lsmeans and multcomp packages were used to conduct a post-hoc analysis [25]. R2 and its confidence interval were calculated for the model, fixed factors, and interactions in linear mixed models using r2glmm package. Mean value and standard deviation (SD) are reported, and the alpha level was set at 0.05.

## 3. Results

### 3.1. Study Flow Diagram

Initially, 27 participants were assessed and randomly assigned to one of two groups (placebo or supplemented). A simple randomization method was chosen. Each subject was assigned to a number that was printed on a separate paper sheet and hidden in an envelope. Then, envelopes were mixed and randomly drawn. Then, numbers were assigned sequentially to the supplemented or placebo group with a 0.9 ratio. Enrolment and randomization were conducted by principal investigator (A.S.S.) who remained blinded to the allocation of participants until formal analysis of results. Three subjects resigned from the supplemented group (*n* = 2 underwent injury related to undergoing physical exercise program applied in a sports team, *n* = 1 resigned without giving a specific reason). Four subjects in total resigned in the placebo group (*n* = 1 due to the injury related to undergoing physical exercise program applied in sports team, *n* = 1 resigned due to acute infection, *n* = 2 resigned from participating in a competitive sport during the trial). Eventually, results from eleven participants in supplemented and nine subjects from the placebo group were analyzed (Figure 1). No important harms or unintended effects were noted in each group.

### 3.2. Participants Examined

Table 2 presents information on the supplemented and control groups before the intervention.

### 3.3. Effects of Supplementation on Response to Physical Exercise Testing

Table 3 presents results of interaction of effects of group (supplemented vs. control) * time (I vs. II vs. III time point) * effects of physical exercise testing on cytokines measured before, just after, and 3 h after physical exercise testing in each time point. None of the examined interactions were statistically significant. 

A statistically significant interaction of the term * group * of exercise response was observed for IL-10 (*p* = 0.01, R2 = 0.09, upper CL = 0.2 lower CL = 0.03). The post-hoc analysis showed significantly higher IL-10 values after the end of the training period 3 h after exercise testing compared to those in the control group, while in the supplemented group, exercise testing did not increase IL-10 3 h after compared to that with rest before. The IL-10 level increased in time point III compared to those in time points I and II in both groups (*p* < 0.05). Figure 2 represents the effects of supplementation and physical exercise testing on IL-10 in the control and supplemented groups. All the results of the post-hoc analysis are listed in Appendix A.

Figure 3 represents the effects of supplementation and physical exercise testing on IL-2/IL-10 in the control and supplemented groups (*p* = 0.09, R2 = 0.05, upper CL = 0.2 lower CL = 0.02).

Table 4 presents results of interaction of effects of group (supplemented vs. control) * time (I vs. II vs. III time point) * effects of physical exercise testing on immunoglobins measured before, just after, and 3 h after physical exercise testing in each time point. A statistically significant interaction was observed for CK MM (*p* = 0.001, R2 = 0.12, upper CL = 0.2 lower CL = 0.05). The post-hoc analysis showed no statistically significant influence of exercise testing on CK MM in the first time point in both groups. On the contrary, significantly higher CK MM values were observed after the end of the training period 3 h after exercise testing compared to that in the control group, while in the supplemented group, exercise testing did not increase CK MM 3 h after compared to that with rest before. The complete results of the post-hoc tests are summarized in Appendix A. The rest of the examined interactions were statistically significant.

Figure 4 represents the effects of supplementation and physical exercise testing on CK MM in the control and supplemented groups (*p* = 0.001, R2 = 0.12, upper CL = 0.25 lower CL = 0.05). None of the interactions were statistically significant.

Table 5 presents results of interaction of effects of group (supplemented vs. control) * time (I vs. II vs. III time point) * effects of physical exercise testing on blood morphology measured before, just after, and 3 h after physical exercise testing in each time point. None of the examined interactions were statistically significant.

## 4. Discussion

### 4.1. Effects of Physical Exercise on the Immune System

In recent years, an increased number of reports have appeared about the influence of exercise on the immune system [2,12,26,27]. While moderate-intensity physical exercise has an immunoprotective effect, intense and/or prolonged exercise may have the opposite effect and contribute to a decrease in immunity [28]. In this aspect, the immune response to physical efforts depends on both the intensity and duration of exercises, as well as the participation of athletes in sports competitions [29,30].

Intense physical exercise is associated with increased anti-inflammatory status and temporary suppression of some immune components [31]. As already mentioned, T lymphocytes play an important role in coordinating the immune response, which according to their phenotypic features resulting from polarization can be divided into type 1 T cells and type 2 regulatory cells. In response to an immune challenge, T cells have the ability to produce cytokines, both pro- and anti-inflammatory. The authors’ own research showed no statistically significant influence of the study date, physical effort, or supplementation on the levels of IL-1α, IL-2, IL-13, and TNF-α. Significant changes were found only in the level of IL-10. In the period of the highest training loads, which fell on the third term of the study, significantly higher levels of this parameter (measured both at rest, exercise, and restitution period) were demonstrated as compared to the values obtained in time points I and II. The above changes concerned both the players from the control and supplemented groups. The results obtained in the presented studies confirmed the previous observations, which indicate that an increase in training loads may modulate changes in immune status by modifying the number of cells and/or cell sensitivity, which results in an enhanced IL-10 response and a subsequently increased immunodepressant effect [10,32].

Moreover, the changes in the level of IL-10 that occurred in the female competitors under the influence of maximal exercise testing until exhaustion are interesting. During the first and second terms of the study, no significant changes in the level of IL-10 measured both immediately after the end of testing and after 3 h of rest were found. On the other hand, in the third term of the study, despite the higher resting values of this parameter compared to those in the previous dates, a statistically significant increase in the level of IL-10 in the restitution period was also demonstrated in relation to the rest values. However, significant changes occurred only in the control group, while in the supplemented group, this increase was statistically insignificant.

### 4.2. Effects of Bovine Colostrum Supplementation on Immune System Function

Supplementation of BC players had an effect on changes in the level of creatine kinase as an indicator of muscle fiber damage (*p* < 0.001). In the second period of the study, statistically significantly lower values of this parameter were shown at rest as compared to that in the control group (Figure 4). On the other hand, in the third period of the study, a statistically significant increase in CK was recorded in the restitution period compared to that in the rest values, but only in the control group. In studies conducted on a group of footballers during the competitive season, it was shown that BC supplementation decreased the CK level 1 day after exercise testing was applied [33]. An important aspect of the observed changes is the reduction of damage to muscle fibers demonstrated after colostrum supplementation, which may shorten the post-exercise recovery time.

The immune system must maintain a delicate balance between immune effector mechanisms and immunoregulatory mechanisms that not only promote immune tolerance and suppress inflammation but also can increase susceptibility to infections. In the first and third terms of the study, the IL-2/IL-10 ratio decreased under the influence of exercise immediately after the end of testing and after 3 h of rest. On the other hand, in the second period of the tests (the transitional period where the players rested), this ratio increased. Although the above observations were not statistically significant, the directions of the observed changes clearly indicate that only during periods of training loads (in our study it was terms I and III), there is a Th1/Th2 balance shift toward the immunosuppressive Th2. Show et al. (2018) [31] believe that the reduction in the number of type 1 T cells in the peripheral blood due to vigorous exercise is likely to be caused by a combination of redistribution to peripheral tissues and polarization promoting a shift toward type 2 T cells, which may impart an inflammatory response to an immune challenge and increase the risk of infection and viral reactivation. Shing et al. (2006) [34] applied an 8-week supply of BC (10 g/day) to cyclists. The authors showed that during the period of intense exercise load, BC supplementation protected athletes against the post-exercise suppression of cytotoxic/suppressor T cell counts (CD3 + 8+), which plays an important role in the cellular resistance to pathogens. Research conducted by the team of Biswas et al. [35] on human peripheral blood mononuclear cells showed that, in the presence of BC, this balance was shifted toward Th1. Similar results were obtained in studies on mice where oral administration of BC was used for 6 months [36]. However, in our own research, no influence of BC supplementation on the Th1/Th2 balance was found. This may be the result of both the small size of the group participating in the research and the dose (3.2 g/day) of the supplement used.

Antibodies (immunoglobulins (Igs)), synthesized and secreted by activated B lymphocytes and their daughter cells called plasmocytes, play a key role in the humoral response. A characteristic property of Igs is, on the one hand, the ability to specifically recognize and bind to the antigen that induced their formation, and on the other hand, the activation of effector functions following the formation of an antigen–antibody complex.

IgG is the main class of Ig found in BC and milk [37]. However, the statistical analysis did not show any significant changes to this parameter in the female athletes under the influence of the effort, timing, and supplementation. Similar results were shown by Carol et al. [38], who used 10 days of BC supplementation in athletes (12.5 g twice a day). In order to increase the stress on the immune system, subjects performed a glycogen-depletion trial the evening before exercise testing. Apart from Igs, the authors also did not show any effect of the given supplement on the level of cytokines, lymphocytes, and neutrophils as well as cortisol levels. Slightly different results were obtained in studies carried out on cyclists, where significant training loads resulted in a reduction in the level of IgG in the subjects, but only in the control group. These changes were not found in competitors supplemented with BC, which according to the authors, may confirm the effectiveness of the supplement in protecting against URTIs [34].

The analysis of the influence of exercise load (intensity, duration of exercise, and frequency) on the level of IgG also does not give unequivocal results. Cordova et al. (2010) [39] conducted research on volleyball players in the period when players are most loaded, namely, in the starting period. Stress tests for refusal were performed by competitors on a bicycle ergometer at the beginning and end of the competition season, which lasted 4 months. In both trials, the athletes had a significant increase in serum IgG and IgM after exercise, along with an increase in the number of circulating lymphocytes, antibody response, and higher cortisol levels. Karacabey et al. (2005) [40] analyzed the influence of aerobic and anaerobic physical exercises on the level of IgG in female athletes and non-training women. The authors showed significantly lower IgG levels in women who did not play sports. Intensive, short-term physical exertion applied in the contestants did not change this parameter, while the aerobic exertion caused an increase in IgG only after 2 and 5 days of exercise.

The analysis of the IGF-I level in the studied players did not increase after BC supplementation. This information is especially important because the World Anti-Doping Agency (WADA) does not recommend using BC due to the possibility of increasing the level of this parameter. The obtained results are consistent with the studies by Davison et al. [41] who used colostrum supplementation in young men. The authors used different doses in their research (from 20 to 40 g/day) and differentiated the duration of supplementation (1 day, 4 weeks, and 12 weeks). There was no increase in IGF-I levels in the supplemented group in any of the above terms. In other studies [42] conducted on athletes who consumed 60 g of BC per day for 4 weeks, no increase in this parameter was also noted.

The analysis of blood morphological parameters concerning the functioning of the immune system did not show the influence of the time of the tests, supplementation, and the applied exercise testing on these parameters. Jones et al. [43] analyzed the effect of 4 weeks of BC supplementation (20 g per day) in physically active people. The authors showed that supplementation had no effect on the level of leukocytes, lymphocytes, monocytes, or neutrophils. However, the changes in these parameters occurred as a result of the physical effort used (2.5 h cycling on a bicycle ergometer with an intensity of 55–60% $\overset{\cdot }{V}O_{2}\max$). Other studies [33] also showed significant changes in the levels of leukocytes, lymphocytes, and granulocytes during exercise, but not with BC supplementation. Siedlik et al. (2016) [44] conducted a meta-analysis of studies in which they assessed the effect of intense physical exercise on the proliferative capacity of peripheral blood lymphocytes. The authors showed that exercise was associated with a slight global inhibition of lymphocyte proliferation, an effect that was more pronounced after exercise for more than an hour. This fact probably explains the lack of statistical differences due to the applied physical effort in the players.

Supplementation with BC is intended to strengthen (or maintain) immune function and reduce the risk of URTIs both after intense and/or prolonged exercise and during the competition, where there is additional mental stress. However, as shown in studies by Brinkworth et al. [17], this supplement shows little efficacy once URTI symptoms are present. Therefore, the use of colostrum supplements should be considered during periods of increased physical activity, as well as in winter periods, where the use of supplementation in physically active people has reduced the number of URTI cases [2]. 

In the above study, dietary supplementation with small doses of BC resulted in post-exercise limitation of muscle fiber damage measured by CK levels and a change in IL-10 levels in response to the physical exercise program. However, the results of the other parameters were not statistically significant. Contrary to our initial hypothesis, no significant effect of colostrum supplementation on the dynamics of the indicators of the immune system functioning in response to the physical exercise tests was found.

The potential limitation of this study is that from the initially calculated sample size (*n* = 24), 20 participants eventually completed the trial and were analyzed. Therefore, the result of this study might be potentially underpowered. Moreover, due to the small sample size and specifications of the study group (female only, young, elite basketball players), one has to be aware of limitations in the generalizability of the trial findings. Further studies should incorporate a larger sample size and compare the effects of supplementation according to physical exercise program characteristics. On the other hand, no harmful effects of BC supplementation were reported during the trial, therefore, BC supplementation seems to be relatively safe in the group free of an allergic reaction to BC. 

The ambiguous results of research on the benefit of BC supplementation may result from the selection of the research group, the period of supply, and the dose, as well as the lack of standardization of this supplement for the presence of active ingredients. It seems that future research should indicate which of the biologically active ingredients should be consumed and in what dose in order to achieve the intended effect of protection against infection. 

## Figures and Tables

**Figure 1 nutrients-13-00118-f001:**
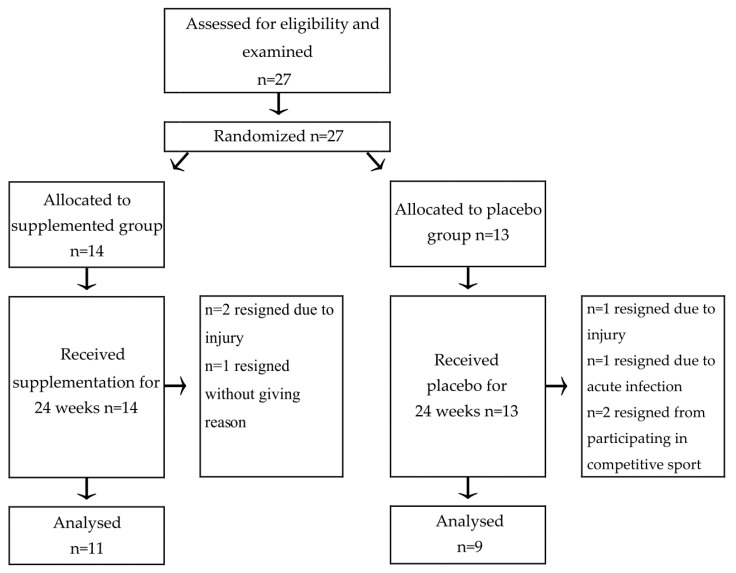
Recruitment process of participants to the trial.

**Figure 2 nutrients-13-00118-f002:**
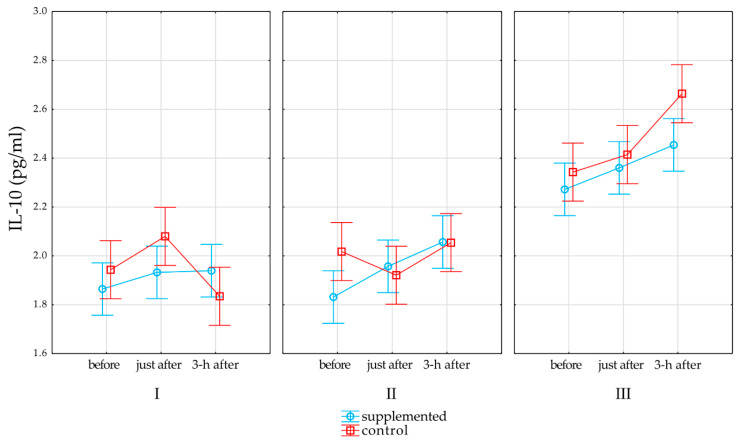
Effects of supplementation on interleukin (IL)-10. Red line denotes changes in the control group, and light blue line denotes changes in the supplemented group. Whiskers denote standard error, while open rectangles for the control group and open circles for the supplemented group denote mean value.

**Figure 3 nutrients-13-00118-f003:**
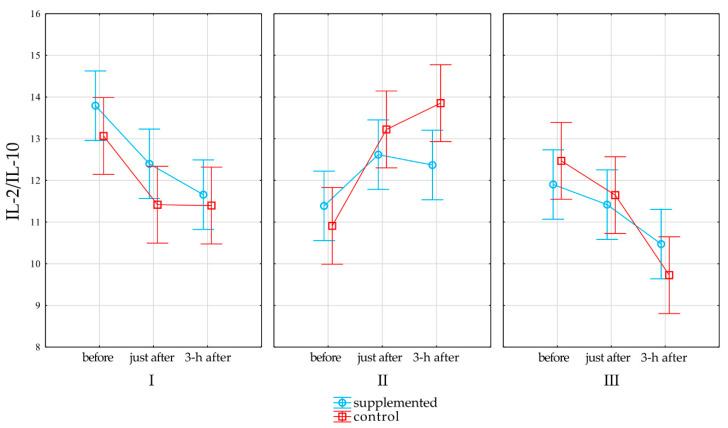
Effects of supplementation on IL-2/IL-10. Red line denotes changes in the control group, and light blue line denotes changes in the supplemented group. Whiskers denote standard error, while open rectangles for the control group and open circles for the supplemented group denote mean value.

**Figure 4 nutrients-13-00118-f004:**
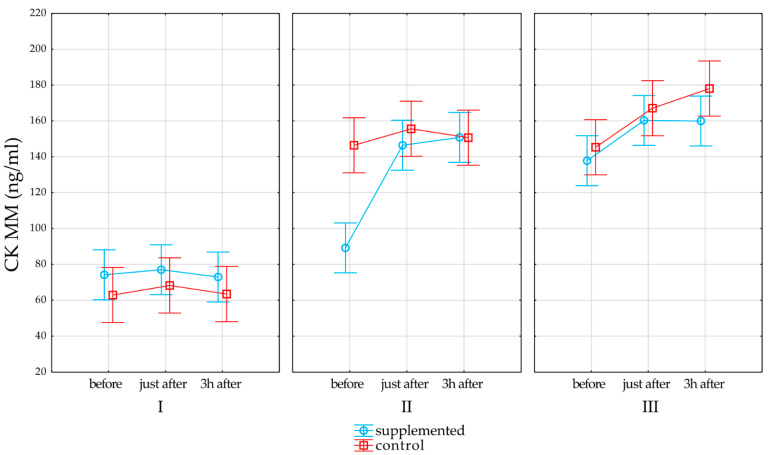
Effects of supplementation on CK MM. Red line denotes changes in the control group, and light blue line denotes changes in the supplemented group. Whiskers denotes standard error, while open rectangles for the control group and open circles for the supplemented group denote mean value.

**Table 1 nutrients-13-00118-t001:** Basic characteristics of the studied groups (mean ± standard deviation (SD)).

Parameter	Group	Time Point I (Mean ± SD)	Time Point II (Mean ± SD)	Time Point III (Mean ± SD)
Time(s)	Control	643.33 ± 96.18	660 ± 107.12	596.67 ± 125
	Supplemented	781.82 ± 126.63	714.55 ± 96.58	774.55 ± 88.81
HRmax (bpm)	Control	192.13 ± 12.29	194.67 ± 7.25	190.9 ± 8.76
	Supplemented	189 ± 18.46	193 ± 6.58	188.5 ± 7.32
Vmax (km/h)	Control	12.85 ± 1.14	12.85 ± 0.98	12.53 ± 0.92
	Supplemented	13.78 ± 1.12	13.03 ± 0.95	13.76 ± 0.84

Time point I–before the intervention, Time point II–in the middle of supplementation, Time point III–after 24 weeks of the intervention, Time–time to peak intensity exercise in seconds; HRmax–maximal heart rate in beats per minute; Vmax–maximal run velocity in kilometers per hour.

**Table 2 nutrients-13-00118-t002:** Basic characteristics of the studied groups (mean ± standard deviation).

Parameters	Supplemented Group (*n* = 11)	Control Group (*n* = 9)
Age (years)	17.09 ± 1.24	16.0 ± 0.67
Body mass (kg)	67.2 ± 6.66	65.6 ± 7.02
Body height (cm)	177.4 ± 5.91	169.5 ± 4.36

**Table 3 nutrients-13-00118-t003:** Effects of supplementation on immune system function indicators.

	Time Point I		Time Point II		Time Point III		
Parameter	Mean ± SD Supplemented Group	Mean ± SD control Group	Mean ± SD Supplemented Group	Mean ± SD Control Group	Mean ± SD Supplemented Group	Mean ± SD Control Group	*p*-Value Interaction Time Point * Group * Effects of Physical Exercise Testing
IL-1alfa (pg/mL) before CPET	34.19 (1.8)	32.76 (1.6)	31.21 (1.8)	31.75 (1.4)	32.58 (1.5)	34.78 (1.4)	0.24
IL-1alfa (pg/mL) just after CPET	33.15 (1.7)	32.89 (1.6)	32.92 (1.3)	32.24 (1.6)	32.85 (1.2)	33.20 (1.5)	
IL-1alfa (pg/mL) 3 h after	32.60 (1.5)	31.94 (1.5)	32.60 (1.3)	32.37 (1.1)	31.29 (1.5)	33.26 (3.8)	
IL-2 (pg/mL) before CPET	25.32 (2.4)	25.34 (1.4)	20.62 (2.2)	22.03 (4.1)	26.96 (2.4)	28.95 (1.7)	0.21
IL-2 (pg/mL) just after CPET	23.78 (1.1)	23.69 (2.1)	24.40 (2.0)	25.26 (1.1)	26.88 (1.4)	28.00 (1.4)	
IL-2 (pg/mL) 3 h after	22.58 (2.2)	20.81 (1.9)	25.27 (2.7)	28.50 (4.7)	25.62 (2.1)	25.65 (2.1)	
IL-10 (pg/mL) before CPET	1.86 (0.3)	1.94 (0.1)	1.83 (0.2)	2.02 (0.2)	2.27 (0.1)	2.34 (0.2)	0.01
IL-10 (pg/mL) just after CPET	1.93 (0.2)	2.08 (0.1)	1.96 (0.2)	1.92 (0.2)	2.36 (0.2)	2.41 (0.2)	
IL-10 (pg/mL) 3 h after	1.94 (0.1)	1.83 (0.2)	2.06 (0.2)	2.05 (0.1)	2.45 (0.2)	2.66 (0.2)	
IL-13 (pg/mL) before CPET	51.36 (4.3)	51.36 (1.8)	51.93 (3.5)	49.11 (2.0)	50.90 (3.0)	53.20 (3.1)	0.21
IL-13 (pg/mL) just after CPET	50.48 (2.1)	49.75 (4.0)	50.23 (2.2)	48.93 (1.8)	53.02 (7.6)	50.36 (2.5)	
IL-13 (pg/mL) 3 h after	48.64 (1.9)	50.43 (2.9)	50.12 (2.1)	49.00 (2.1)	49.07 (2.6)	49.12 (3.3)	
TNF-α (ng/L) before CPET	2.58 (0.2)	2.76 (0.4)	2.90 (0.4)	3.41 (0.8)	1.66 (0.2)	1.82 (0.2)	0.13
TNF-α (ng/L) just after CPET	3.04 (0.5)	2.69 (0.6)	3.33 (0.6)	3.37 (0.2)	1.92 (0.3)	2.31 (0.5)	
TNF-α (ng/L) 3 h after	2.62 (0.4)	2.84 (0.3)	3.05 (0.3)	3.23 (0.2)	1.69 (0.2)	2.01 (0.3)	
IL-2/IL-10 before CPET	13.79 (1.9)	13.07 (0.8)	11.39 (1.9)	10.91 (1.7)	11.90 (1.3)	12.47 (1.5)	0.09
IL-2/IL-10 just after CPET	12.40 (1.2)	11.42 (1.2)	12.62 (2.0)	13.22 (1.0)	11.42 (0.7)	11.65 (1.0)	
IL-2/IL-10 3 h after	11.66 (1.1)	11.40 (1.1)	12.37 (1.5)	13.85 (2.0)	10.47 (0.9)	9.73 (1.5)	
IL-10/TNF-α before CPET	0.73 (0.1)	0.71 (0.1)	0.64 (0.1)	0.63 (0.2)	1.39 (0.1)	1.31 (0.2)	0.16
IL-10/TNF-α just after CPET	0.65 (0.1)	0.80 (0.1)	0.60 (0.1)	0.57 (0.1)	1.25 (0.2)	1.09 (0.2)	
IL-10/TNF-α3 h after	0.76 (0.2)	0.65 (0.1)	0.68 (0.1)	0.64 (0.1)	1.47 (0.2)	1.34 (0.2)	

Time point I–before the intervention, Time point II–in the middle of supplementation, Time point III–after 24 weeks of the intervention, CPET-cardiopulmonary exercise testing, IL–interleukin, TNF–tumor necrosis factor.

**Table 4 nutrients-13-00118-t004:** Effects of supplementation on immune system function indicators.

	Time Point I		Time Point II		Time Point III		
Parameter	Mean ± SD Supplemented Group	Mean ± SD Control Group	Mean ± SD Supplemented Group	Mean ± SD Control Group	Mean ± SD Supplemented Group	Mean ± SD Control Group	*p*-Value Interaction Time Point * Group * Effects of Physical Exercise Testing
IgG (g/L) before CPET	11.81 (3.6)	13.92 (6.1)	11.49 (3.5)	14.99 (4.8)	11.40 (4.4)	12.88 (4.2)	0.8
IgG (g/L) just after CPET	13.92 (2.6)	13.85 (4.4)	14.21 (2.3)	13.63 (3.4)	14.35 (3.9)	12.23 (2.0)	
IgG (g/L) 3 h after CPET	16.61 (3.2)	13.69 (4.5)	16.75 (3.2)	13.55 (3.4)	18.07 (4.0)	11.81 (2.3)	
IGF-1 (ng/mL) before CPET	258.18 (71.5)	268.37 (47.7)	241.69 (77.8)	227.93 (29.9)	435.87 (15.8)	424.27 (16.6)	0.59
IGF-1 (ng/mL) just after CPET	277.07 (66.9)	232.06 (25.2)	277.01 (72.4)	260.26 (37.1)	443.70 (17.2)	428.15 (10.0)	
IGF-1 (ng/mL) 3 h after CPET	277.84 (39.7)	261.23 (32.1)	245.28 (37.0)	248.18 (111.9)	433.21 (19.7)	419.73 (14.3)	
CK MM (ng/mL) before CPET	74.16 (16.7)	62.88 (6.9)	89.18 (33.5)	146.39 (53.4)	137.78 (31.7)	145.29 (23.2)	0.001
CK MM (ng/mL) just after CPET	77.00 (18.8)	68.23 (7.9)	146.42 (25.0)	155.60 (26.4)	160.27 (15.3)	167.09 (20.6)	
CK MM (ng/mL) 3 h after CPET	72.92 (15.5)	63.43 (7.0)	150.79 (21.7)	150.65 (20.0)	159.94 (14.3)	178.05 (17.7)	

Time point I–before the intervention, Time point II–in the middle of supplementation, Time point III–after 24 weeks of the intervention, CPET-cardiopulmonary exercise testing, IgG-immunoglobulin G, IGF-1-insulin-like growth factor 1.

**Table 5 nutrients-13-00118-t005:** Effects of supplementation on immune system function indicators.

	Time Point I		Time Point II		Time Point III		
Parameter	Mean ± SD Supplemented Group	Mean ± SD Control Group	Mean ± SD Supplemented Group	Mean ± SD Control Group	Mean ± SD Supplemented Group	Mean ± SD Control Group	*p*-Value Interaction Time Point * Group * Effects of Physical Exercise Testing
WBC (10^3^/µL) before CPET	5.35 (0.7)	6.33 (1.5)	7.18 (1.1)	6.86 (1.1)	6.32 (1.1)	7.23 (1.6)	0.14
WBC (10^3^/µL) just after CPET	10.95 (2.1)	10.10 (1.8)	10.97 (1.5)	12.04 (2.7)	10.38 (1.7)	11.57 (3.6)	
WBC (10^3^/µL) 3 h after CPET	7.57 (1.4)	7.13 (0.9)	8.10 (1.4)	7.93 (1.5)	7.78 (0.6)	7.98 (1.3)	
LYM (10^3^/µL) before CPET	1.81 (0.3)	1.83 (0.4)	2.05 (0.5)	2.18 (0.4)	2.04 (0.6)	2.12 (0.4)	0.11
LYM (10^3^/µL) just after CPET	4.29 (0.6)	3.88 (0.5)	3.85 (0.5)	4.32 (0.7)	3.95 (0.6)	3.88 (0.9)	
LYM (10^3^/µL) 3 h after CPET	1.74 (0.3)	1.76 (0.4)	1.95 (0.4)	1.96 (0.3)	1.86 (0.5)	1.93 (0.2)	
MON (10^3^/µL) before CPET	0.38 (0.1)	0.39 (0.1)	0.50 (0.1)	0.53 (0.1)	0.43 (0.1)	0.50 (0.1)	0.97
MON (10^3^/µL) just after CPET	0.74 (0.2)	0.63 (0.2)	0.73 (0.1)	0.69 (0.2)	0.70 (0.2)	0.70 (0.3)	
MON (10^3^/µL) 3 h after CPET	0.41 (0.1)	0.37 (0.1)	0.41 (0.1)	0.38 (0.1)	0.45 (0.1)	0.50 (0.1)	
GRA (10^3^/µL) before CPET	3.15 (0.6)	4.11 (1.5)	4.61 (0.9)	4.12 (1.0)	3.88 (0.7)	4.61 (1.3)	0.18
GRA (10^3^/µL) just after CPET	5.94 (1.4)	5.59 (1.3)	6.43 (1.2)	6.99 (2.1)	5.74 (1.2)	6.99 (2.6)	
GRA (10^3^/µL) 3 h after CPET	5.41 (1.4)	5.00 (0.9)	5.77 (1.2)	5.62 (1.3)	5.49 (1.0)	5.56 (1.2)	

Time point I–before the intervention, Time point II–in the middle of supplementation, Time point III–after 24 weeks of the intervention, WBC–*white blood cell*, *LYM*–lymphocyte, MON–monocyte, GRA–granulocyte.

## Data Availability

The data presented in this study are available on request from the corresponding author. The data are not publicly available due to sport club internal rules.

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
