# Peer review of "Effects of Long-Term Supplementation of Bovine Colostrum on the Immune System in Young Female Basketball Players. Randomized Trial"

_nutrients, 2020, doi:10.3390/nu13010118_

Round 1
Reviewer 1 Report
Thank you for the opportunity to review this manuscript. I have some comments on the overall study and then suggestions on specific areas of the manuscript. These comments/suggestions are only what in my opinion will help improve the reporting of your study. This study adds to the literature by investigating long term supplementation of bovine colostrum and recruiting athletes from team sports. These are two areas which have not received much attention in previous studies. The key limitations of the study are the small sample size and acute exercise bouts that are likely to be too short for an intervention like bovine colostrum to be needed or at least for you to be able to show the impact of the intervention on immunity.
Abstract: I would suggest switching the order of the two background sentences in your abstract and in terms of bovine colostrum re-word to clarify that bovine colostrum may be a countermeasure to exercise-induced immune dysfunction. I appreciate that we are always limited by word count when it comes to an abstract. If possible, however, I would look to expand on the detail of the methods and the results in your abstract. For example, for the methods I suggest specifying age of participants (i.e. what do you mean by young), athlete level of participants (e.g. elite, olympic or university/college), the dose and frequency of the bovine colostrum supplementation, duration and nature of exercise tests, list of blood measures examined. For the results, I felt there was too much emphasis on the statistical significance of the results without any information to the reader on the actual data (e.g. magnitude of change, difference between groups). One of the reasons why I have these suggestions is that the abstract can be (for some readers) the only aspect of the study that is read. Finally, in terms of your conclusion, could it be that the dose of bovine colostrum was not sufficient to impact on immunity? It also worth being specific on your measures of immunity n your conclusion (e.g. plasma immunoglobulins and cytokines)
Introduction:
Lines 58 - 73. I would suggest re-writing this section, particularly considering two things: check the terms used (e.g. replace use of vague text, "selective mechanisms") and ensure text is supported by citations.
Lines 74-98. Please strengthen the rationale for use of bovine colostrum supplementation in team sports. This should include reference to previous studies of the demands of team sports on immunity and infection risk.
Methods:
Participants/Figure 3: What the sample size based on? Did you have any sample size calculation. If not, I would ensure that you report this limitation in the methods. For example, it may be that your study was underpowered to detect the impact of bovine colostrum. Can you also provide further information on your participants please (e.g training hours at point of study)? Also, was this a full squad, members of different squads? Did you have any exclusion criteria?
Physical performance: I would provide the mean duration and standard deviation of each maximum stress test in this section. This allows the reader to examine whether the exercise was strenuous enough. In Table 1, what unit is time measured in? Seconds? Please specify, with a suggestion for this to be in minutes.
Material examination: You have IgF, this should be IGF-1?
Statistical analysis: Please consider providing more detail on the approach here. You have three factors (group, time, point) in your study design but yet only provide one p value per parameter in each of your tables. I'm unsure that your analysis allows you (particularly due to the small sample size) to investigate impact of bovine colostrum on resting and exercise-induced immunity. What if you analysed the impact of bovine colostrum vs placebo on resting measures at baseline, midway and 24 weeks in a two way mixed model? I then consider having colostrum vs placebo and before, just after and 3 hour exercise response at Stage II (another two way mixed ANOVA) and then same again at stage III (another two way mixed ANOVA). Given your sample size, this may be informative. I would also consider reporting effect sizes in your tables as your analyses may be underpowered to detect a statistically significant difference, hence concluding there is no effect may be misleading. I don't think you need a comparison of groups at Stage I? Or perhaps move to supplementary material?
Discussion: I feel that this section is far too long for the current study. Please shorten and ensure the order of the text aids the reader (e.g. main findings, comparison to previous studies, strengths/limitations, implications for future research, conclusion). I think one key thing that the authors need to consider is whether there was sufficient scope for bovine colostrum to have an impact given that it is an intervention which is likely most useful under the most strenuous training phases or prolonged exercise. There was tendency for speculation on other immune parameters or background information which would be more suitable for an introduction section.
I hope you find these comments useful.
Author Response
Response to Reviewer 1 Comments
Thank you very much for the comments on our paper entitled „ Effects of long-term supplementation of bovine colostrum on the immune system in young female basketball players. Randomized trial”. The manuscript has been revised following all the remarks of the expert referees.
Reviewer 1
( ) I would not like to sign my review report
(x) I would like to sign my review report
English language and style
(x) Extensive editing of English language and style required
( ) Moderate English changes required
( ) English language and style are fine/minor spell check required
( ) I don't feel qualified to judge about the English language and style
|
Yes |
Can be improved |
Must be improved |
Not applicable |
|
|
Does the introduction provide sufficient background and include all relevant references? |
( ) |
(x) |
( ) |
( ) |
|
Is the research design appropriate? |
( ) |
( ) |
(x) |
( ) |
|
Are the methods adequately described? |
( ) |
( ) |
(x) |
( ) |
|
Are the results clearly presented? |
( ) |
( ) |
(x) |
( ) |
|
Are the conclusions supported by the results? |
( ) |
( ) |
(x) |
( ) |
Comments and Suggestions for Authors
Thank you for the opportunity to review this manuscript. I have some comments on the overall study and then suggestions on specific areas of the manuscript. These comments/suggestions are only what in my opinion will help improve the reporting of your study. This study adds to the literature by investigating long term supplementation of bovine colostrum and recruiting athletes from team sports. These are two areas which have not received much attention in previous studies. The key limitations of the study are the small sample size and acute exercise bouts that are likely to be too short for an intervention like bovine colostrum to be needed or at least for you to be able to show the impact of the intervention on immunity.
Re: We would like to give many thanks for this kind review and Reviewer’s effort.
Abstract: I would suggest switching the order of the two background sentences in your abstract and in terms of bovine colostrum re-word to clarify that bovine colostrum may be a countermeasure to exercise-induced immune dysfunction. I appreciate that we are always limited by word count when it comes to an abstract. If possible, however, I would look to expand on the detail of the methods and the results in your abstract. For example, for the methods I suggest specifying age of participants (i.e. what do you mean by young), athlete level of participants (e.g. elite, olympic or university/college), the dose and frequency of the bovine colostrum supplementation, duration and nature of exercise tests, list of blood measures examined. For the results, I felt there was too much emphasis on the statistical significance of the results without any information to the reader on the actual data (e.g. magnitude of change, difference between groups). One of the reasons why I have these suggestions is that the abstract can be (for some readers) the only aspect of the study that is read. Finally, in terms of your conclusion, could it be that the dose of bovine colostrum was not sufficient to impact on immunity? It also worth being specific on your measures of immunity n your conclusion (e.g. plasma immunoglobulins and cytokines)
Re: All comments and suggestions of the Reviewer regarding the abstract were taken into account.
Introduction:
Lines 58 - 73. I would suggest re-writing this section, particularly considering two things: check the terms used (e.g. replace use of vague text, "selective mechanisms") and ensure text is supported by citations.
Re: Those lines have been changed and supplemented with new references, in accordance with the Reviewer's comments.
Lines 74-98. Please strengthen the rationale for use of bovine colostrum supplementation in team sports. This should include reference to previous studies of the demands of team sports on immunity and infection risk.
Re: In the given lines, the justification for the use of colostrum in team games has been added.
Methods:
Participants/Figure 3: What the sample size based on? Did you have any sample size calculation. If not, I would ensure that you report this limitation in the methods. For example, it may be that your study was underpowered to detect the impact of bovine colostrum.
Re: We are really grateful for this suggestion. Sample size cauclulation was added to the description to paragraph „4.1. Participants”. Limitation of the study was added to one before last paragraph of discussion.
Can you also provide further information on your participants please (e.g training hours at point of study)? Also, was this a full squad, members of different squads? Did you have any exclusion criteria?
Re: Information was added to description of participants in materials and methods. All the players were one team and were subject to the same training regime, namely: training sessions took place every day in the afternoon and lasted from 1.5 to 2 hours
Physical performance: I would provide the mean duration and standard deviation of each maximum stress test in this section. This allows the reader to examine whether the exercise was strenuous enough. In Table 1, what unit is time measured in? Seconds? Please specify, with a suggestion for this to be in minutes.
Re: Table 1 legend has been extended. In addition, parameters obtained in subsequent time points (Time point II I Time point III) were added.
Material examination: You have IgF, this should be IGF-1?
Re: Acronym was changed, many thanks for pointing this out.
Statistical analysis: Please consider providing more detail on the approach here. You have three factors (group, time, point) in your study design but yet only provide one p value per parameter in each of your tables. I'm unsure that your analysis allows you (particularly due to the small sample size) to investigate impact of bovine colostrum on resting and exercise-induced immunity. What if you analysed the impact of bovine colostrum vs placebo on resting measures at baseline, midway and 24 weeks in a two way mixed model? I then consider having colostrum vs placebo and before, just after and 3 hour exercise response at Stage II (another two way mixed ANOVA) and then same again at stage III (another two way mixed ANOVA). Given your sample size, this may be informative.
Re: We appreciate this insightful comment. In fact, we have run analysis as You suggested. We have attached excel spreadsheet with results in the above version of manuscript. We agree that such models would be a realy useful way of our data presentation. We will publish those results in further papers. However, the primary outcome of the above study is to examine effect of supplementation on response to physical exercise test, therefore all levels of those three factors should be incorporated into analysis.
I would also consider reporting effect sizes in your tables as your analyses may be underpowered to detect a statistically significant difference, hence concluding there is no effect may be misleading.
Re: Many thanks for this suggestion, R2 was calculated and added to results section
I don't think you need a comparison of groups at Stage I? Or perhaps move to supplementary material?
Re: We are grateful for this suggestion. Indeed, we agree that comparison at Stage I is not needed. However, p-value from interaction time point*group*effects of physical exercise testing is calculated based on data from all three stages, therefore We would prefer to leave it in one table with other stages, to not potentially induce confusion.
Discussion: I feel that this section is far too long for the current study. Please shorten and ensure the order of the text aids the reader (e.g. main findings, comparison to previous studies, strengths/limitations, implications for future research, conclusion). I think one key thing that the authors need to consider is whether there was sufficient scope for bovine colostrum to have an impact given that it is an intervention which is likely most useful under the most strenuous training phases or prolonged exercise. There was tendency for speculation on other immune parameters or background information which would be more suitable for an introduction section.
Re: Discussion was revised. Is it suitable in Reviewers opinion in the current version?
I hope you find these comments useful.
Reviewer 2 Report
STRUCTURE
The manuscript is NOT properly structured:
- Introduction
- Materials and Methods
- Results
- Discussion
- Conclusions
- References
INTRODUCTION
- State specific objectives.
- No hypotheses are included.
MATERIAL AND METHODS
- Line 290: indicate from which year
- Description of trial design (such as parallel, factorial) including allocation ratio is missing
- Were there any important changes to methods after trial commencement (such as eligibility criteria)?
- Eligibility criteria for participants?
- Line 323: another table 2?
- Line 324: another table 1?
- You have already put table 1 and table 2 in other sections
- How sample size was determined?
- Method used to generate the random allocation sequence?
- Indicate type of randomisation; details of any restriction (such as blocking and block size)
- Mechanism used to implement the random allocation sequence? Describe any steps taken to conceal the sequence until interventions were assigned
- Who generated the random allocation sequence, who enrolled participants, and who assigned participants to interventions?
- Figure 3. Recruitment process of participants to the trial à it must be in results
RESULTS
- Participant flow. For each group, the numbers of participants who were randomly assigned, received intended treatment, and were analysed for the primary outcome. For each group, losses and exclusions after randomisation, together with reasons.
- Were there any important harms or unintended effects in each group?
DISCUSSION
- Summarise key results with reference to study objectives.
- Line 227: What do you think this might be due to?
- Trial limitations, addressing sources of potential bias, imprecision, and, if relevant, multiplicity of analyses?
- Generalisability (external validity, applicability) of the trial findings?
- Interpretation consistent with results, balancing benefits and harms, and considering other relevant evidence?
CONCLUSIONS
- There are no conclusions, there must be
REFERENCES
- References follow the style indicated.
OTHER INFORMATION
- Registration number and name of trial registry?
Author Response
Response to Reviewer 2 Comments
Thank you very much for the comments on our paper entitled „ Effects of long-term supplementation of bovine colostrum on the immune system in young female basketball players. Randomized trial”. The manuscript has been revised following all the remarks of the expert referees.
Reviewer 2
I would not like to sign my review report
( ) I would like to sign my review report
English language and style
( ) Extensive editing of English language and style required
( ) Moderate English changes required
(x) English language and style are fine/minor spell check required
( ) I don't feel qualified to judge about the English language and style
|
Yes |
Can be improved |
Must be improved |
Not applicable |
|
|
Does the introduction provide sufficient background and include all relevant references? |
( ) |
(x) |
( ) |
( ) |
|
Is the research design appropriate? |
( ) |
(x) |
( ) |
( ) |
|
Are the methods adequately described? |
( ) |
(x) |
( ) |
( ) |
|
Are the results clearly presented? |
( ) |
(x) |
( ) |
( ) |
|
Are the conclusions supported by the results? |
( ) |
(x) |
( ) |
( ) |
Comments and Suggestions for Authors
STRUCTURE
The manuscript is NOT properly structured:
- Introduction
- Materials and Methods
- Results
- Discussion
- Conclusions
- References
Re: The structure of the manuscript has been revised as recommended by the Reviewer. Is it suitable in Reviewers opinion in the current version?
INTRODUCTION
- State specific objectives.
- No hypotheses are included.
- Re: Specific objectives and hypotheses are included in the manuscript.
MATERIAL AND METHODS
- Line 290: indicate from which year
- • Re: the above information has been completed in the text.
- Description of trial design (such as parallel, factorial) including allocation ratio is missing
- Re: We are really grateful for this suggestion. These descriptions were added to the description to paragraph „4.1. Participants”.
- Were there any important changes to methods after trial commencement (such as eligibility criteria)?
- Re: No changes to the test methods were made after the start of the study.
- Eligibility criteria for participants?
- The research covered players playing in one team, playing games in the first league and extra-class, and agreed to participate in the study - all respondents had valid medical examinations allowing them to participate in training and competitions.
- Line 323: another table 2?
- Re: The order of the table has been corrected.
- Line 324: another table 1?
- Re: Table order has been corrected.
- You have already put table 1 and table 2 in other sections
- Re: The order of the tables in the manuscript has also been corrected.
- How sample size was determined?
- Re: We are really grateful for this suggestion. Sample size calculation was added to the description to paragraph „4.1. Participants”.
- Method used to generate the random allocation sequence?
- Re: Many thanks for suggestions about randomization. Randomization has been described in the first paragraph of results. Each subject was assigned to number was printed on a separate paper sheet and hidden in envelope. Then, envelopes were mixed and randomly drawn.
- Indicate type of randomisation; details of any restriction (such as blocking and block size)
- Re: Simple randomization method was chosen
- Mechanism used to implement the random allocation sequence? Describe any steps taken to conceal the sequence until interventions were assigned
- Re: Each subject was assigned to number was printed on a separate paper sheet and hidden in envelope. Then, envelopes were mixed and randomly drawn. Then, numbers were assigned sequentially to supplemented or placebo group with 0.9 ratio.
- Who generated the random allocation sequence, who enrolled participants, and who assigned participants to interventions?
- Re: Enrolment and randomization was conducted by principal investigation (A.S.S.) who remained blinded to allocation of participants until formal analysis of results
- Figure 3. Recruitment process of participants to the trial à it must be in results
- Re: Figure 3 was moved to the results
RESULTS
- Participant flow. For each group, the numbers of participants who were randomly assigned, received intended treatment, and were analysed for the primary outcome. For each group, losses and exclusions after randomisation, together with reasons.
- Re: Figure 3 was moved to the results and described.
- Were there any important harms or unintended effects in each group?
- Re: Information was added to description in the first paragraph in results section.
DISCUSSION
- Summarise key results with reference to study objectives.
- Re: The above suggestion has been entered into the manuscript.
- Line 227: What do you think this might be due to?
- Re: It seems that the reason for the lack of differences between the supplemented group and the control group in the analyzed Th1 / Th2 balance may be both the small size of the group and the first dose of the supplement used. The above information was hidden in the text.
- Trial limitations, addressing sources of potential bias, imprecision, and, if relevant, multiplicity of analyses?
- Re: Trial limitations description was added to one but last paragraph in discussion
- Generalisability (external validity, applicability) of the trial findings?
- Re: Trial generalisability description was added to one but last paragraph in discussion
- Interpretation consistent with results, balancing benefits and harms, and considering other relevant evidence?
- Re: information was added to last sentence in one but last paragraph in discussion
CONCLUSIONS
- There are no conclusions, there must be
- Re: Conclusions resulting from the work are included in the text.
REFERENCES
- References follow the style indicated.
OTHER INFORMATION
- Registration number and name of trial registry?
- Re: Unfortunately, the trial was not registered in any database. Further trials would be registered in clinicaltrials.gov or similar database to be accessible for viewers.
Round 2
Reviewer 2 Report
No further comments.